# Counseling in Vape Shops: A Survey of Vape Shop Managers in Switzerland

**DOI:** 10.3390/ijerph182010861

**Published:** 2021-10-15

**Authors:** Sandra Joss, Anna Moser, Julian Jakob, Kali Tal, Jean-François Etter, Kevin Selby, Anna Schoeni, Philippe Poirson, Reto Auer

**Affiliations:** 1Institute of Primary Health Care (BIHAM), University of Bern, 3012 Bern, Switzerland; sandra.joss@biham.unibe.ch (S.J.); anna.moser@students.unibe.ch (A.M.); julian.jakob@biham.unibe.ch (J.J.); kali.tal@biham.unibe.ch (K.T.); anna.schoeni@biham.unibe.ch (A.S.); 2Department of Pediatrics, University Hospital Bern, Inselspital, 3010 Bern, Switzerland; 3Institute of Global Health, Faculty of Medicine, University of Geneva, 1211 Geneva, Switzerland; jean-francois.etter@unige.ch; 4Center for Primary Care and Public Health (Unisanté), 1011 Lausanne, Switzerland; kevin.selby@unisante.ch; 5Sovape, 1211 Geneva, Switzerland; p.poirson@sovape.fr

**Keywords:** e-cigarette, smoking cessation, vape shops

## Abstract

Vaporizers (e-cigarettes) are the most common smoking cessation aids in Switzerland, but we do not know what information customers receive in vape shops. We surveyed vape-shop managers to find out what recommendations they make to their customers. An interdisciplinary group developed the questionnaire. Respondents self-reported their smoking history, demographics, and the recommendations they thought they would give to hypothetical customers in clinical vignettes. We also queried if they collaborated with health care professionals. Of those contacted, 53.8% (70/130) of vape-shop managers responded, and 52.3% (68/130) were included in the final analysis. Managers were mostly male and ex-smokers who switched to vaporizers; 60.3% would encourage a hypothetical smoker with high nicotine dependence to start with the highest possible nicotine concentration when switching to vaporizers. For this smoker, 36.9% would recommend high (≥15 mg/mL), 32.3% medium (6–14 mg/mL), and 3.1% low (1–5 mg/mL) nicotine concentrations. The rest adapted their recommendations to fit the customer or device; 76.5% reported that physicians referred customers to them, and 78.8% would attend a course given by experts in the field of vaporizers and smoking cessation. Vape-shop managers varied widely in the recommendations they gave customers. Most reported ongoing collaboration with health care professionals and were interested in improving their counselling skills through training.

## 1. Introduction

Vaporizers, also called electronic nicotine delivery systems (ENDS) or e-cigarettes, are the most common smoking cessation aids in Switzerland [1]. Vaporizers are battery-powered devices that heat a liquid, or e-liquid. These liquids are usually made from propylene glycol and glycerol, flavorings, and nicotine. Most consider them a less harmful alternative to combustible tobacco cigarettes and mounting evidence suggests vaporizers that contain nicotine are effective smoking cessation aids [2,3]. The attractiveness of vaporizers as smoking cessation aids is increased by the variety of devices and types and flavors of e-liquids but customers may be confused by the many options.

In Switzerland and elsewhere, consumers can learn about vaping and vaping equipment at specialized stores called “vape shops”. In addition to giving customers advice about choosing vaporizers and e-liquids, research from the USA and UK suggests that many shops advise smokers on how to quit smoking [4,5,6,7,8,9]. Vape shop employees personally tailor the advice they give smokers about choosing a device and offer continuing practical help [10]. Smokers who purchased their first vaporizers in a vape shop and received professional advice from vape shop employees may accomplish high and stable smoking cessation rates [11]. Vape shop customers reported in another study that they received information about smoking cessation and reduction [8].

Vape shops have the potential to become partners in more formal smoking-cessation efforts. Most vape-shop managers are ex-smokers themselves, and many share personal experiences with customers along with information they found on the internet, but few employees are formally trained to counsel smokers who want to quit [4,5,12,13]. In France, vape-shop managers can get certified through a special training [14]. In the UK, the National Health Service lists e-cigarettes as an option for smoking cessation and suggests that smokers get advice from vape shops [15]. Such forms of training or collaborations with health institutions do not exist in Switzerland. A lack of training may lead to vape-shop managers giving customers outdated advice on vaping, nicotine dependence and smoking cessation, and may have difficulty understanding the information they have read [7,9]. They may also overestimate the safety of vaporizers; for example, vape-shop managers in Los Angeles thought vaporizers were safer than nicotine replacement therapy (NRT) [16]. Collaborations between vape shops and health professionals active in smoking cessation may aid smokers who want to quit, but only if the advice given by vape shops is sound.

In Switzerland, we know little about the advice vape shops provide, or the effects of that advice on the behavior of customers who want to quit smoking tobacco cigarettes. We need to determine which devices and nicotine concentrations vape-shop managers recommend to customers, what they tell customers about the risks and benefits of vaping, and if they give other advice, e.g., to set a quit date, and whether they work with health care professionals.

We thus surveyed vape-shop managers in Switzerland, presenting them with hypothetical customer vignettes and questions that captured their advice to customers and their interactions with health care professionals. We further surveyed them about their interest in improving their counselling skills through training. 

## 2. Materials and Methods

### 2.1. Participants and Procedures

We sent invitations to all vape shops with a physical address in the German- and French-speaking parts of Switzerland. The ‘ARPV’ (Association Romande des Professionnels de la Vape) sent us their list of members and we obtained a list of ‘SVTA’ (Swiss Vape Trade Association) members from their website. We identified vape shops that belonged to neither association via online searches (keywords: ‘E-Zigarette’, ‘e-cigarette’, ‘elektronische Zigarette’, ‘cigarette électronique’, ‘vape’ and ‘Dampf’) on online phone books (local.ch, search.ch) and on Google Maps, in May and June, 2020. We excluded shops that did not specialize in vaporizers (e.g., kiosks and petrol stations), shops that only operated online or sold e-liquids but not hardware. We identified 130 “brick-and-mortar” shops with a valid postal address and mailed them a questionnaire by post. We also invited them to participate via an email that included a link to an online version of the questionnaire (hosted on Survey Monkey). The questionnaire was in French and German (translated from an English original). Both the ARVP and SVTA reached out to their members, informing them of the survey on their websites and in closed Facebook groups. We sent two e-mail reminders and one postal reminder to non-responders. Data was collected from August through September 2020. We requested each vape shop select only one person to complete the survey, preferably the owner or manager. Two vape shop mangers were involved in the early development of the survey.

### 2.2. Survey Development and Content

To develop the survey, we drew on the literature and worked with vape-shop managers and vapers to design the questions and three hypothetical customer vignettes. Researchers experienced in smoking cessation reviewed survey questions for face validity and exhaustiveness. The first vignette elicited vape-shop managers’ recommendations for type of device and nicotine concentration for a smoker with high nicotine dependence (nine points in the Fagerström Test for Nicotine Dependence) who wanted to switch to a vaporizer. They could choose between four types of devices: (1) disposable ‘ciga-likes’ that cannot be refilled or recharged (first generation); (2) refillable and rechargeable ‘vape pens’ that usually have adjustable airflow but a set temperature (2nd generation); (3) customizable ‘box mods’, that often carry two batteries and are the most advanced devices on the market (3rd generation); and (4) pod systems (‘pods’), which are rechargeable devices with exchangeable pre-filled liquid-cartridges (fourth generation) [17]. The second vignette was designed to determine if vape-shop managers could identify a case of e-liquid intolerance that caused asthma symptoms and the third determined what vape-shop managers would tell a customer with stable coronary artery disease about the risks posed by vaporizers (for details on the vignettes, see Appendix A).

The questionnaire also determined if vape-shop managers supported smokers trying to quit by (1) encouraging them to set a quit date, (2) telling them about nicotine withdrawal symptoms, (3) assessing their degree of nicotine dependence, and (4) asking them about their smoking history and attempts to quit. We asked managers if health care providers referred smokers to their vape shop, and if they recommended customers visit health care providers. We asked managers if they were interested in a training course for their employees. We also asked them if they thought vaporizers and other nicotine-containing products were harmful. Respondents indicated their age, sex, education, smoking, and vaping history.

The questionnaire contained single- and multiple-choice questions and the following optional open-ended (free text) questions: “How do you assess nicotine dependence?” “What additional advice would you give [the hypothetical customer in vignette one] to successfully quit smoking?” “What else is important when counseling a customer for the first time?” Since answers to the last two questions were similar, we grouped them together for the analysis. The full questionnaire is presented in the Appendix A.

### 2.3. Statistical Analysis

We reported means and standard deviation for continuous variables, and proportions for categorical variables. We analyzed available data and did not impute missing data. (See the Appendix A for missing values.) When respondents checked “refuse to answer” or if they failed to answer a question, we categorized the answer as missing. We used STATA 15.1 for our analyses (StataCorp, College Station, TX, USA). We thematically analyzed free text responses to open-ended questions and inductively identified patterns as they emerged. Two team members (S.J. and A.M.) familiarized themselves with the qualitative data, coded it, and extracted initial themes, then reviewed and refined the themes. We translated selected free-texts from French (F) or German (G) to English to highlight thematic field

## 3. Results

Of the 130 vape shops we invited, representatives of 70 shops (53.8%) answered the survey. We excluded data from two shops: one answered all questions with “refuse to answer” and another one answered less than 50% of the questionnaire. Our final analysis included 68/130 (52.3%) participants. Appendix A lists the number of missing values for each question. Most participants were managers and/or owners of the shop. Eight participants indicated they were staff. One refused to answer. Most respondents (81.5%, 53/65) were men; mean age was 40 years (SD 10); 29.0% (18/62) had a tertiary education degree. More than 80% (54/66) were former smokers who switched to vaporizers. Of the included shops, 75.4% (49/65) also sold devices online and only 9.0% (6/67) reported selling tobacco products. See Table 1 for the characteristics of vape-shop managers.

### 3.1. Hypothetical Customer Vignettes

#### 3.1.1. Vignette One—Smoker with High Nicotine Dependence

Faced with a highly dependent hypothetical smoker who wanted to switch to a vaporizer, 29.4% (20/68) recommended starting with a vape pen (second generation), 29.4% (20/68) a box mod (3rd generation), 29.4% (20/68) or a pod system (fourth generation), and 4.4% (3/68) a ciga-like (first generation); 7.4% did not recommend a specific device (5/68). More than a third of respondents (36.9%, 24/65) recommended the customer select a nicotine concentration of 15 mg/mL or more, 32.3% (21/65) suggested between 6 and 14 mg/mL, and 3.1% (2/65) suggested between 1 and 5 mg/mL. No respondent recommended nicotine-free liquids; 20.0% (13/65) did not make a general recommendation. Most respondents (60.3%, 41/68) would encourage this customer to start with the highest possible nicotine concentration; 8.8% (6/68) would encourage him to start with the lowest nicotine concentration. Table 2 lists the results of Vignette One.

#### 3.1.2. Vignette Two—Customer with Intolerance to E-Liquid

Most respondents (62.1%, 41/66,) answered adequately to the second clinical vignette: 39.4% (26/66) said the asthma symptoms were likely caused by intolerance to the e-liquid; 22.7% (15/66) advised the customer to consult a physician. Over a third (37.9%, 25/66) did not answer adequately: 15.2% (10/66) said the symptoms were caused by the high concentration of nicotine and 12.1% (8/66) said they were caused by the propylene glycol. 

#### 3.1.3. Vignette Three—Customer with History of Coronary Artery Disease

Most respondents (78.4%, 51/65) would tell a smoker with a history of coronary artery disease that vaporizers are not risk free but are better than conventional cigarettes after a heart attack (see Appendix A). Fewer (13.8%, 9/65) asserted that vaporizers did not increase risk of another heart attack and that the customer could use them safely. Two respondents (3.1%, 2/65) answered that vaporizers increase heart attack risk as much as conventional cigarettes. Most (44.4%, 28/63) would give a current smoker with a history of heart attack the same advice as they gave any other customer; 15.9% (10/63) recommended a vaporizer with a low nicotine concentration and 11.1% (7/63) would suggest the customer first discuss it with his physician.

### 3.2. Collaboration with Health Care Professionals

Most respondents (76.5%, 52/68) said that some customers were referred to the shop by physicians. Referrals from other health care professionals such as nurses or pharmacists were less common (see Figure 1 for details). Half of respondents (50.0%, 34/68) said they recommend customers visit a physician; 30.9% (21/68) said they never referred customers to health care professionals.

### 3.3. Topics Addressed When Counselling Customers

The vast majority of respondents (82.4% (56/68) stated that they give advice about smoking cessation. Most respondents (64.7%, 44/68) reported they never encouraged customers to set a quit date and that they always (44.1%, 30/68) or often (25.0%, 17/68) assessed their customers’ smoking history. They tell customers always (45.6%, 31/68) or often (25.0%, 17/68) about nicotine withdrawal symptoms (see Figure 2).

### 3.4. Assessment of Nicotine Dependence

More than half the respondents always assessed their customers’ nicotine dependence (Figure 2). We analyzed 46 text answers that explained how they assessed nicotine dependence. Over half said they asked about the number of cigarettes customers smoked each day (63.0%, 29/46) and which brand of cigarettes they smoked (34.8%, 16/46). A few (17.4%, 8/46) also said they ask customers how long they are awake before they smoke their first morning cigarette. Few (15.2%, 7/46) asked about prior cessation attempts (6%, 3/46) or the duration of tobacco addiction. Some respondents (13.0%, 6/46) said they let customers try different nicotine concentrations in the vaporizer instead of specifically assessing their degree of dependence. 

### 3.5. Perceived Harmfulness of Nicotine-Containing Products and Interest in Participating in A Course

When asked to rank the harmfulness of nicotine-containing products on a scale (0 = no harm, 9 = extremely harmful), respondents ranked cigarettes as most harmful and vaporizers as least harmful; 9.2% (6/65) reported that vaporizers did not harm consumers (Appendix A). On average, respondents ranked NRT as slightly more harmful than vaporizers.

Most (78.8%, 52/66) managers would participate in a course for vape shop employees; and 10.6% (7/66) had already attended such a course. 

### 3.6. Additional Advice Given to Customers

We thematically analyzed 49/54 answers to the free text question, “What else is important when counseling a customer for first time?” We also analyzed 61/62 answers to the question, “What additional advice do you give him (hypothetical customer in vignette one) to successfully quit smoking?” Three topics emerged from our analysis. The first topic was about materials: vape-shop managers emphasized the importance of each customer finding the right vaporizer and e-liquid flavor. 

One respondent explained: “The choice of the device takes a central place in the advice because it will directly influence the flavors and the way in which nicotine is delivered. But above all it will replace the “gesture of the smoker”, it is therefore necessary that the device is appealing and accepted by the customer. The choice of liquid, its flavor, is essential in the cessation of smoking…” F1. 

They taught customers how to maintain the device and use it safely. One respondent wrote, “…To avoid leaving his electronic cigarette to the reach of children and store it well away from sources of heat. To turn off his cigarette when he no longer uses it…” F11.

The second topic was nicotine: vape shop owners said they gave customers general information about nicotine and described nicotine withdrawal and nicotine overdose symptoms. One respondent wrote: “*It should be pointed out that the nicotine does not have the same fast absorption time when vaporizing…*”. G27. Another explained: “*Initially, the nicotine content should not be too low, so that you do not have to reach for a cigarette right away. After that (when used to it) slowly reduce.*” G67.

The third topic was behavioral advice and support: respondents shared the kind of advice they gave smokers to help them successfully quit, including identifying and avoiding smoking triggers, how to break habits, and motivational advice. Some encouraged customers to make a follow-up visit. One respondent wrote, ”*…in case of a failure, if the person still smokes, we encourage them to come back and to discuss and re-motivate…*” F35.

## 4. Discussion

Vape-shop managers in Switzerland were typically ex-smokers who switched to vaporizers (81.8%). They recommended a variety of devices for vaping. Most (60.3%) recommended smokers with high nicotine dependence start vaping at the highest possible nicotine concentration. Most (78.4%) would tell a smoker that nicotine-containing vaporizers are not risk free but are safer than conventional cigarettes; 9.2% of managers thought vaporizers were completely safe. Managers reported helping customers find the device, flavour, and nicotine concentration that was right for them. Managers reported giving behavioural advice for smoking cessation. They also said they educate customers about maintenance and safety of vaporizers. Most managers (76.5%) had customers who were referred to their shop by a physician. A minority of vape-shop managers (10.6%) had attended a course on vaping and smoking cessation for vape shop employees, but most (78.8%) would be interested in attending such a course. 

There was no consensus among managers on the best device for smoking cessation, perhaps because the efficacy of many smoking cessation devices sold in vape shops is unclear. Nicotine delivery is likely more efficient in later generations of the devices than in first generation devices [18]. Some managers in our study still recommended a first generation device. Their varied recommendations might reflect the rapid evolution of the market, their different perceptions of the effectiveness of particular devices, and how much they think a device might appeal to consumers. Whatever the device, most managers recommended smokers start with a high nicotine concentration, and this aligns with physicians’ recommendations for pharmacologic nicotine replacement therapy.

Vape-shop managers seem to be in favor of smoking cessation with the help of vaporizers. Only 9% of the participating shops in our study sold tobacco products. This might stand for the idea that most vape-shop managers want their customers to get off tobacco completely. Previous research on vape shops did report concerns about regulations of vaping products: they fear that restriction of the types of devices would lead to “less promise for aiding in cessation” [19].

As former smokers who have switched to vaporizers, vape-shop managers relied on their personal experience to advise smokers who want to use vaporizers to help them quit. Prior UK and USA studies also found that vape-shop managers were usually former smokers who used vaporizes themselves [7,9,10,16]. Vape-shop managers explain how they successfully switched from smoking to vaping and help customers individually find the right combination of device, e-liquid flavor, and nicotine concentration. Finding the right combination is associated with smoking cessation [20]. One study described vape-shop managers as “experts by experience” [10]. Another compared the advice given by vape shop employees to peer support for smoking cessation [13]. This peer support might be one of the reasons why smokers had high and stable smoking cessation rates in a study by Polosa et al. [11]. Smoking cessation stakeholders should consider policies that integrate vape shops into smoking cessation efforts, making use of the “peer-supported smoking cessation counseling” they provide because vape shops may attract smokers who might not seek help with smoking cessation within the health care system. 

Few vape-shop managers were specifically trained to counsel smokers, so they may be unaware of the risks of vaping. Some managers in Switzerland thought that vaporizers were completely safe, which aligns with the findings of studies conducted in vape shops in Los Angeles and New York [16,21]. For example, in Los Angeles only 25% of vape shop employees were aware of any study that described the risks of vaping [9]. Many vape-shop managers may have difficulty interpreting research findings [7]. Since vape-shop managers are offered no official training in Switzerland, they likely obtain their information from personal experience and online sources, as other studies reported [5,8]. More than a third of the respondents in our survey could not adequately handle a customer with intolerance to e-liquid. This finding highlights that some vape-shop managers lack proper training to counsel smokers who want to quit with the help of vaporizers. Other researchers have expressed concern about smoking advice provided by vape shops, especially when employees lack proper training [22]. Since managers have a financial interest in encouraging smokers to vape, they may have incentive to downplay risks. If managers are provided with balanced information and evidence-based training, supplemented by educational material like brochures and videos, customers will benefit. At best, vape shop employees should undergo standardized training before counseling customers. Such standardized training could, for example, be developed by the vape community itself in collaboration with smoking-cessation professionals. A certification like the one that exists in France could serve as a model [14]. This could help health care professionals to identify reputable vape shops to refer clients for product support and advice. Such partnerships are known in the UK [15]. In return, vape shop owners could educate health care professionals about vaping and about their customers’ preferences in vaping devices and e-liquids. 

A collaboration between vape shops and health care professionals would increase the likelihood that smokers who want to switch to vaporizers are properly counselled. Though earlier studies found only a few physicians recommended vaporizers for smoking cessation [23,24], the majority of vape-shop managers in our study reported that physicians and other health care professionals referred customers. Vape shops seem interested in working more closely with health care professionals [10], and physicians who know more about vaping might be more likely to recommend vaporizers to reduce harm in smokers for whom NRT or behavioural therapy are not working. 

## 5. Conclusions

Vape-shop managers made widely varied recommendations to smokers who wanted to switch to vaporizers. Since closer collaboration between vape-shop managers and health care professionals might benefit smokers who want to quit, and most vape-shop managers would attend educational courses on vaping and smoking cessation, we suggest policy makers and health care professionals look for opportunities for mutual information exchange. 

## 6. Limitations

Our results are based on the self-reports of vape-shop managers, so they may not reflect the advice they would give to real, rather than hypothetical customer. We recommend future studies include observations of interactions between shop managers and customers to confirm these self-reports. Two vape-shop managers were involved in the early development of the survey. The link to the survey was posted online by the vape trade associations, thus we cannot exclude that those two managers have participated. This may have biased the results. We did not translate the survey to Italian—our study was limited to vape shops in the German- and French-speaking parts of Switzerland—so our results might not reflect practices in the Italian-speaking part of Switzerland. Future research should include all regions of Switzerland. Finally, our moderate participation rate (54%) raises the chance that this study may have suffered from selection bias. Vape-shop managers who are interested in research and invested in smoking cessation may have completed the survey in greater numbers than less interested. 

## Figures and Tables

**Figure 1 ijerph-18-10861-f001:**
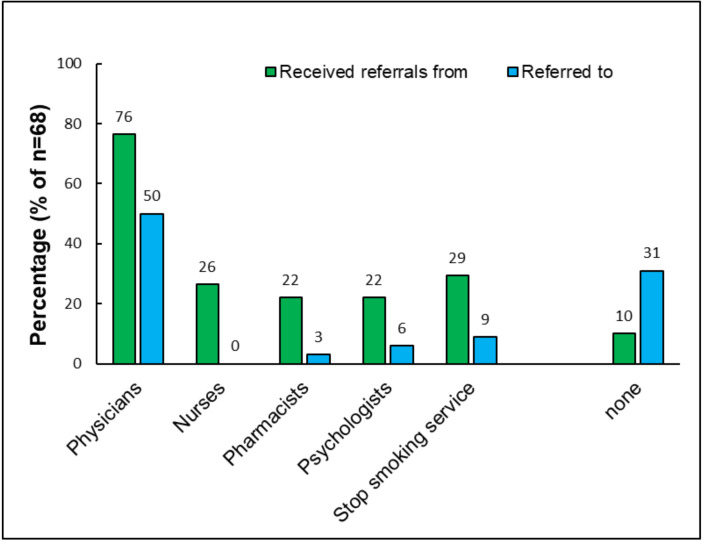
Percentage of respondents who say that health care professionals refer customers and who say they refer customers to health care professionals.

**Figure 2 ijerph-18-10861-f002:**
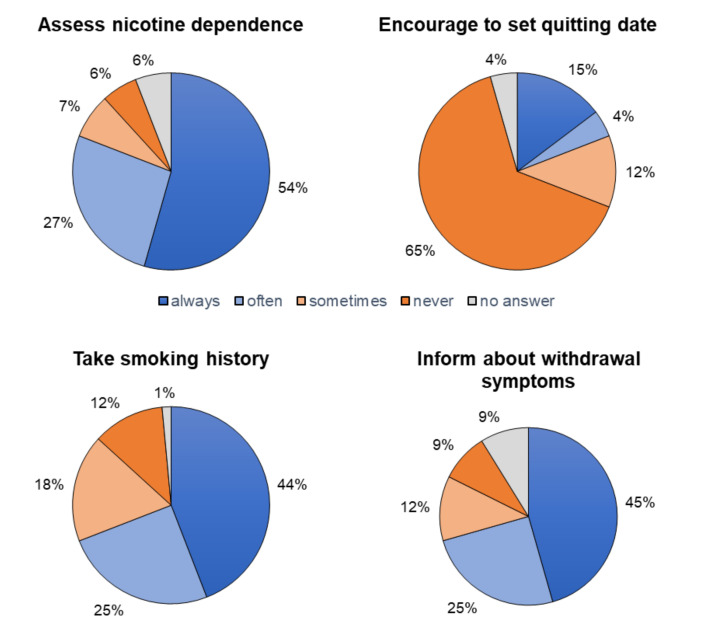
Topics addressed when counselling customers self-reported reported by respondents. *n* participants = 68.

**Table 1 ijerph-18-10861-t001:** Respondent and shop’ characteristics.

CHARACTERISTIC (*n* = 68) *	*n* or mean	% or SD
Sex		
Female	12	18.5
Male	53	81.5
Mean age (years)	40	10
Language		
German	32	47.1
French	36	52.9
Smoking/Vaping status		
Ex-smoker, using vaporizer	54	81.8
Dual user	7	10.6
Using vaporizer, never smoked	1	1.5
Ex-smoker, not using vaporizer	3	4.6
Never smoked, never used vaporizer	1	1.5
Role in shop		
Owner ^a^	34	50.7
Manager	25	37.3
Staff	8	11.9
Highest level of education		
Mandatory school	4	6.5
Apprenticeship	28	45.1
High school	12	19.4
Tertiary level	18	29
Selling tobacco products		
Yes	6	9
No	61	91
Selling online		
Yes	49	75.4
No	16	24.6
Experience working in vape shop (years)	4.3	2.6

* Incomplete demographic data in seven participants; ^a^ six owners were also managers, two other owners were also staff.

**Table 2 ijerph-18-10861-t002:** Results of Case Vignette One (recommendations to a current smoker with high nicotine dependence interested in quitting).

	*%* (*n*)
First choice of device	
Ciga-like	4.4% (3)
Pod system	29.4% (20)
Vape pen/eGo	29.4% (20)
Box Mod	29.4% (20)
No general recommendation	7.4% (5)
Nicotine concentration	
0 mg/mL	0.0% (0)
1–5 mg/mL	3.1% (2)
6–14 mg/mL	32.3% (21)
≥15 mg/mL	36.9% (24)
Depending on device	20.0% (13)
Depending on customers’ preferences	7.7% (5)
Recommendation to start with:	
Highest possible nicotine concentration	60.3% (41)
Lowest possible nicotine concentration	8.8% (6)
No recommendation	30.9 % (21)
Additional NRT	
Yes	9.4% (6)
No	90.6% (58)

Abbreviation: NRT Nicotine replacement therapy.

## Data Availability

The data underlying this article will be shared on reasonable request to the corresponding author.

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
