# Peer review of "Counseling in Vape Shops: A Survey of Vape Shop Managers in Switzerland"

_ijerph, 2021, doi:10.3390/ijerph182010861_

Round 1

Reviewer 1 Report

Paper is very well designed and presented. It helps address important questions about the current relatively disorganized vape shop component of the e-cigarette industry. I recommend publication as is.

Reviewer 2 Report

Counseling in vape shops: a survey of vape shop managers in Switzerland.

Review:

I appreciate the opportunity to review this manuscript. Overall, the paper is well written and contributes to the scientific literature specific to understanding the role of vape shop employees on smoking behavior in Switzerland. I present my comments and suggestions below for each section of the manuscript.

Abstract:

  • It would be helpful to have the analytic sample size somewhere in this section.

Introduction:

The Introduction reads well but could use more background, such as what is already known about the impact of vape shop employees on consumer behavior. This could strengthen the argument that understanding vape shop employees’ practices is needed. The citation below might be helpful

  • Polosa R, Caponnetto P, Cibella F, Le-Houezec J. Quit and Smoking Reduction Rates in Vape Shop Consumers: A Prospective 12-Month Survey. International Journal of Environmental Research and Public Health. 2015; 12(4):3428-3438. https://doi.org/10.3390/ijerph120403428

Furthermore,

  • Page 2, line 57: The word “strengths” here does not make sense to me.

Methods:

  • It is unclear whether the managers/vapers that helped designed the survey were also allowed participated in the study. If so, this may bias study results.

Results:

Overall, the results are well written given that there is a mix of quantitative and qualitative findings. However, there are a lot of results, some of which I believe need to be discussed (see my comments in the next section).

  • Referring to all employees as “managers” is confusing given that “role in shop” (see Table 1) clearly categorizes employees into three groups. I recommend using the word “employees” to refer to respondents, or even “respondents” might work.

Discussion:

  • It seems like one of the takeaways of this study is that training for vape shop employees is necessary to protect consumers who are switching to ENDS for smoking cessation. Results suggest that many recommendations are based on anecdotal and not empirical evidence, and you discuss that policymakers should incorporate policies into smoking cessation programs that include vape shop employees as resources. Consider also mentioning the importance of standardized training with these points.
  • I do not see any discussion about results involving the second vignette. It appears that intolerance to e-liquid is a subject not many vape shop employees fully understand/can make proper recommendations for. These results may also support standardized training for vape shop employees.
  • Given that only 10% of the stores that participated sell tobacco products, do you think there is a chance that employees are pro-smoking cessation via ENDS since they only sell ENDS? Consider commenting on this in the discussion. This thought already aligns with one point that you make on page 8, lines 281-282 about the financial implications of suggestions made by vape shop employees.
  • Another point is that studies have shown that vape shop employees report concerns about ENDS regulation, which could be another reason why they support ENDS as a smoking cessation tools. That is, in theory, if ENDS are deemed more beneficial than harmful at the population level, regulations may be less strict. Here’s the citation:
    • Berg CJ, Barker DC, Sussman S, et al. Vape Shop Owners/Managers’ opinions about FDA regulation of e-cigarettes. Nicotine Tob. Res. 2020;23(3):535-542. doi:10.1093/ntr/ntaa138
  • I assume the last paragraph is the limitations paragraph, but please make this clearer by either creating a subheading or starting the paragraph with an additional sentence stating that the following sentences include study limitations. Also, great mention of likely self-selection bias due to only about half the vape shop employees electing to participate.

Tables 1-2:

  • Consider reporting decimals to the tenth place for percentages and standard deviations.

Reviewer 3 Report

The paper deals a relevant task for understanding the dynamics of the smokers. The paper could be considered for publication after few changes. Mainly, the authors should show the situation in the other European countries for comparing with the Swiss situation. Simultaneously, they should show the situation worldwide. In literature there ar different papers dealing this subject, so it should be easy for the authors catch this information.
